# Testing of a Polarimetric Current Sensor in the Frequency Domain

**DOI:** 10.3390/s21093008

**Published:** 2021-04-25

**Authors:** Sławomir Andrzej Torbus, Jacek Andrzej Michalski

**Affiliations:** 1Institute of Mathematics, Kazimierz Wielki University in Bydgoszcz, Powstańców Wielkopolskich 2, 85-090 Bydgoszcz, Poland; 2Faculty of Civil Engineering, Mechanics and Petrochemistry, Warsaw University of Technology, Łukasiewicza 17, 09-400 Płock, Poland; jacek.michalski@pw.edu.pl

**Keywords:** frequency analysis, Discrete Fourier Transform, polarimetric current sensor, single mode optical fiber, measurement error, total harmonic distortion

## Abstract

This paper proposes an original model of a polarimetric current sensor, in which the measuring coil was made of a single mode telecommunication optical fiber ITU-T G.652, G.653, G.655, and G.657. This sensor was subjected to the commercialization process, which was carried out by a company combining the functionality of a technology transfer center with the capabilities of the Startit Fund Sp. z o.o. The published results included the analysis of the implementation readiness, the analysis of the market potential, the valuation of the industrial property rights of the invention and indicated further directions of scientific research on the sensor, which include the frequency analysis of measurement signals. This prompted the conduct of relevant scientific research. In this paper, the idea of measurement of current using polarimetric current sensor with optical fiber coil has been briefly characterized. It shows the definition and basic properties of the Discrete Fourier Transform (DFT). It discusses the technique of determining the value of each harmonic of signal at the input and output of polarimetric current sensor. The value of measurement errors and total harmonic distortion (THD) have been calculated. The general conclusions for disturbances in the processing realized in polarimetric current sensor have been formulated. In addition, the impact of the molar concentration of the dopant GeO_2_ in the core of the single mode telecommunication optical fibers and the impact of the number of turns of the measuring coil on the distortion accompanying the process of processing have been determined. Therefore, it can be concluded that the key result obtained during the research is the confirmation of the fact that single mode telecommunication optical fibers can be used to build the measuring coil of a polarimetric sensor used for measuring alternating currents. This means that the considered sensor, when measuring this type of currents, does not introduce additional distortions and distortions of their waveforms.

## 1. Introduction

One of the types of fiber optic sensors are polarimetric sensors of the magnetic field, in particular current sensors with a fiber optic measuring coil, which are based on the magnetooptical Faraday effect.

Their characteristic feature is that they enable the determination of the instantaneous value of the current intensity on the basis of a non-electrical signal. This property is called the non-invasiveness of the optical fiber [1], because without interfering with the power grid infrastructure, it is possible to measure the current in phase conductors of power lines. It is not possible if you use classic current transformers, which, despite galvanic separation of the measuring circuit from the main circuit, give the output an electrical signal of a small value, proportional to the measured current, which can be measured with available measuring instruments.

Polarimetric current sensors with a fiber-optic measuring coil, in relation to classic measuring transformers, show [2,3]: high accuracy and dynamic range, a wide frequency range, from the constant component to the hundredth harmonic, low mass and dimensions, very good shock resistance, ease and safety of installation, no saturation, galvanic isolation from the high voltage power line, and no interference with the external environment.

The first records of this issue date back to the 1980s [4]. However, the following years brought stagnation, which could be caused by high costs of the sensor implementation compared to a classic current transformer and the lack of appropriate optical fibers. Return to work on the issue of polarimetric sensors can be observed from the end of the 1990s [5]. The active element of the polarimetric sensor can be a crystal or an optical fiber [4,6,7] through which laser light propagates. Under the influence of an external magnetic field, the angle of polarization (the angle of rotation of the plane of polarization), propagated in the active element of light, is changing.

Many scientific studies have considered sensors with a fiber optic measuring coil made of single mode [8,9,10,11,12] or multimode optical fibers [13,14]. Sensors of this type, compared to those in which the active element is a crystal [6,7], can be much easier mounted on the poles of the existing low, medium, and high voltage power lines in order to implement remote measurement and control-protection systems. However, in the published scientific papers, a certain gap can be observed: in the proposed solutions, commonly known telecommunications fiber solutions are not considered (in accordance with the recommendation numbers of the International Telecommunications Union (ITU-T) called G.652, G.653, and G.655), as well as their latest solutions—bending resistant optical fibers (in accordance with the recommendation number of the International Telecommunications Union (ITU-T) are called G.657) and their dedicated wavelengths of light, i.e., 1310 nm and 1550 nm. It is these types of single mode telecommunication optical fibers, characterized by a low price and the above-mentioned wavelengths of light, whose generators (light sources) are widely available, as elements that can be used to build polarimetric current sensors with a fiber-optic measuring coil. These problems have already been considered in the authors’ works [15,16,17,18,19,20,21,22,23].

The issues of designing and using polarimetric current sensors in metrology are currently very popular and are the subject of theoretical and experimental research [3,9,24,25,26,27,28,29,30,31,32,33]. It should be added here that there are also studies on the design of single-mode telecommunication optical fibers for other purposes, with particular emphasis on the effect of the light wavelength and the molar concentration of the GeO_2_ impurity in the core of this type of fiber on the metrological properties of polarimetric current sensors and the possibility of their use in remote systems measurement, control, and protection of power lines.

The issues related to the design and manufacture of polarimetric current sensors, as well as the implementation of remote measurement and control-protection systems based on them, are the subject of research carried out in laboratories of large companies from the protection power industry and several academic centers around the world.

Dynamic changes in the power industry are constantly observed, and an example can be the requirements set by the IEC 61,850 [26,34] standard, which says that current transformers used to measure and test the operating states of power lines will have to work on the basis of low-voltage signals and digitally communicate with the systems EPAP (Electrical Power Automation Protection) and CaSS (Control and Supervision System) systems.

In Poland, polarimetric current sensors are not widely used, despite the fact that research and development work has been carried out for over two decades [1]. Classic transformers are still used, and the new solutions work only alongside conventional ones. However, in countries such as the USA, Canada, Great Britain, Sweden, France, and Finland [1], where expenditure on research and implementation is greater, and the economy is focused on environmental protection polarimetric current sensors have already been used in power stations and substations for the last few years.

Finally, it is worth noting that the original sensor model proposed in the literature [15,16,17,18,19,20,21,22,23], was subject to a commercialization process carried out by a company combining the functionality of a technology transfer center with the fund’s capabilities investment project Startit Fund Sp. z o.o. The published results included the analysis of the implementation readiness, the analysis of the market potential, and the valuation of the industrial property rights of the invention called “Polarimetric current sensor with a measuring coil made of a single mode telecommunications fiber” [35]. The currently conducted scientific works result directly from the indications contained in the results of commercialization, which take into account the following aspects:design of optical fibers for Faraday sensors;accuracy of current measurement with Faraday sensors determined on the basis of the classical error theory, the theory of measurement uncertainty and the frequency analysis of measurement signals;remote sensors and measurement systems.

## 2. The Idea of Operation of a Polarimetric Current Sensor

Polarimetric current sensor—the detailed design of which, and properties resulting from the use of the optical fiber were shown in papers [16,19,20,36]—is used to determine current flowing in a power line conductor based on the following parameters: light polarization angle, measurement wavelength, Verdet constant, and number of turns of the measuring coil.

The optical fiber is an optically inactive body. This means that, in absence of an external magnetic field, the angle of polarization of light propagated through the waveguide is not modified. On the occurrence of an external magnetic field, the waveguide becomes optically active: the angle of polarization of propagated light changes (as a result of the magnetooptic Faraday effect) as described by equation [37]:(1)α=V·L·B
where: α—polarization angle [rad], V—Verdet constant (proportionality factor) $\left[\frac{\mathrm{rad}}{T\cdot m}\right]$, L—path on which light interacts with the magnetic field [m], and B—magnetic field induction [T].

The value of the measured current using the polarimetric current sensor (Figure 1) depends on the following factors: number of turns of the measuring coil, light polarization angle determination accuracy, and properties of the waveguide material. It can be described by the following equation [19]:(2)$$I=\frac{\alpha }{\mu_{0}\cdot V\cdot N}$$
where: $\mu_{0}=4\cdot \pi \cdot 10^{-7}$
$\left[\frac{V\cdot s}{A\cdot m}\right]$—magnetic permeability of vacuum and N—number of turns of the measuring coil.

The value of current *I* occurring in Equation (2) can represent the constant value of constant current and the instantaneous value of alternating current. The analyzer enables measurement of instantaneous values of waveforms as a function of time, providing the light polarization angle value as a function of time (instantaneous value). The variant is noteworthy when the measurement is given a sinusoidal alternating current with two polarities—positive and negative. If the light propagated in the optical fiber propagates in the direction (parallel) to the direction of the magnetic field (rotation to the right) set for the positive polarity of the sinusoidal waveform, then the analyzer will show a positive value of the polarization angle, otherwise, negative.

This model is the basis for the proprietary sensor model, in which the measuring coil is made of different standards of single mode telecommunications optical fibers: three standards of not resistant to bending ITU-T G.652 (4.582 M% of GeO_2_), G.653 (9.344 M% of GeO_2_), G.655 (7.075 M% of GeO_2_), and two standards of resistant to bending ITU-T G.657A (3.617 M% of GeO_2_) and G.657B (2.970 M% of GeO_2_). This design of the measurement coil is an innovative approach, and the use of bending-resistant optical fibers allows the construction of measurement coils with a small radius. Additionally, this solution can use lasers with commonly known optical wavelengths—1310 nm and 1550 nm.

## 3. Definition and Selected Properties of the Discrete Fourier Transform

The Discrete Fourier Transform (DFT) is a mathematical procedure used for signal analysis in the frequency domain. This is how we can determine the content of harmonics in the signal being analyzed. They can be defined as a discrete series of output components X(m) in the frequency domain [38]. In exponential and trigonometric forms, it takes the form:(3)$$\begin{array}{l} X\left(m\right)=\sum_{n=0}^{N-1}x\left(n\right)\cdot e^{-j\cdot \frac{2\cdot \pi \cdot n\cdot m}{N}}=\\ =\sum_{n=0}^{N-1}x\left(n\right)\cdot \left(\cos\left(\frac{2\cdot \pi \cdot n\cdot m}{N}\right)-j\cdot \sin\left(\frac{2\cdot \pi \cdot n\cdot m}{N}\right)\right) \end{array}$$
where: x(n)—discrete series of values in the time domain of variable continuity x(t), $j=\sqrt{-1}$—imaginary unit, *n*—input sample index in the time domain (*n* = 0, 1, 2, 3,…, *N*−1), *m*—index of DFT output samples in the frequency domain (*m* = 0, 1, 2, 3,…, *N*−1), *N*—number of samples in the input series, and number of points in the DFT output series.

The frequency determination method is described in [38]. Frequency values of subsequent *N* points on the frequency axis, on which DFT lines are determined, can be determined as follows:(4)$$f_{analisys}\left(m\right)=\frac{m\cdot f_{s}}{N}$$
where: $f_{s}$—frequency of sampling of the original (continuous) signal [Hz].

The Discrete Fourier Transform features symmetry. This means that if we determine an *N*-point DFT for the true input series, we will obtain *N* individual complex output values of DFT, of which only the initial $\frac{N}{2}$ are independent. Therefore, to obtain DFT for signal x(n), it is enough to compute $\frac{N}{2}$ of values X(m), where $0\leq m\leq \frac{N}{2}-1$ [38]. The output values ranging from $X\left(\frac{N}{2}\right)$ till X(N−1) of DFT contain no additional information on the spectrum of the true series x(n) [38]. In addition, we can see the following dependency [15]:(5)$$\begin{array}{l} X\left(N-m\right)=\sum_{n=0}^{N-1}x\left(n\right)\cdot e^{j\cdot \frac{2\cdot \pi \cdot n\cdot m}{N}}=\\ =\sum_{n=0}^{N-1}x\left(n\right)\cdot \left(\cos\left(\frac{2\cdot \pi \cdot n\cdot m}{N}\right)+j\cdot \sin\left(\frac{2\cdot \pi \cdot n\cdot m}{N}\right)\right) \end{array}$$

If we check into Equation (5), we can notice that component X(N−m) differs only in the sign from that contained in Equation (3) defining component X(m), these are coupled components. This means that, in terms of modulus, the components are equal to each other and they differ only in phase.

Another very important feature of DFT is its linearity, which means that DFT of a sum of signals is equal to the sum of transforms of each signal [38]. Owing to this property, we can analyze complicated signals represented as sum of sinusoidal signals: distorted cyclic waveforms and not just a signal representing a single sinusoidal waveform.

After determining DFT values of components, we can establish on this basis amplitudes and true RMS values of the harmonic components of the signal being subject to frequency analysis. If the input waveform contains a constant component, the following equation obtained on the basis of information contained in the work [38], takes the form:(6)$$X_{0}=\frac{X\left(0\right)}{N}$$
where: X(0)—amplitude of the output DFT constant component.

On the other hand, effective values of the sinusoidal components of the input waveform will be described by the following formula [38]:(7)$$X_{m}=\frac{2\cdot X\left(m\right)}{N\cdot \sqrt{2}}$$
where: X(m)—output amplitude of DTF output stripes and *m* = 1, 2, 3,…, $\frac{N}{2}-1$.

Given the symmetry of DFT, note that by determining the *N*-point DFT of the input signal, we obtain only information about $\frac{N}{2}$ of the components with indices ranging from 0 to $\frac{N}{2}-1$, where index 0 corresponds to the constant component and indices ranging from 1 to $\frac{N}{2}-1$ correspond to individual sinusoidal components.

Accordingly, the true RMS value of the signal subjected to DFT will be described by the following equation [39], in accordance with the Parseval’s theorem:(8)$$X_{RMS}=\sqrt{X_{0}^{2}+\sum_{m=1}^{\frac{N}{2}-1}X_{m}^{2}}$$

Based on the obtained DFT results the relative harmonic content (total harmonic distortion, THD) can be determined. It is defined as follows [39]:(9)$$THD=\frac{\sqrt{\sum_{m=2}^{\frac{N}{2}-1}X_{m}^{2}}}{X_{1}}\cdot 100\%$$
where: *X*_1_—harmonic with index 1 (so-called basic harmonic).

## 4. Frequency Analysis of Periodic Output Signal from the Polarimetric Current Sensor and Input Signal Subjected to Processing

The sensors considered in the experiment consisted of a light source (single mode laser from EXFO, model FLS-2600 with linear light polarization, with relative power level adjustable from 0.0 to 6.0 dBm with a 0.1 dBm step, enabling light wave length selection from within the 1518.00–1568.00 nm range with 0.01 nm step (the 1550.00 nm light wave length at the relative power level of the laser equal to 4.0 dBm was used for the measurements), with FC/PC interfaces, single mode telecommunications fiber (conforming to ITU-T standards G.652, G.653, G.655, G.657A and G.657B), a fiber optic polarimeter (from Agilent, model 8509B, without internal laser light source, with FC/PC interfaces), and a PC (with dedicated software for registration of instantaneous values of light polarization angle). See Figure 2 for the diagram of the measuring system.

Before proceeding with the measurements, while the strength of measured current was 0 A, the polarimeter has been calibrated, so that the power of linearly polarized light on its input was maximal (the best visibility within the optical system).

In addition, current clamps (from HEME, model PR1030) and a Rogowski coil (from FLUKE, model i2000 FLEX) were installed on the tested wire with the current for the measurement of the true RMS value of the current. Both the meters were connected to two identical digital multimeters (from EnergyLab, model EM5512) measuring the true RMS value. The current clamps were calibrated without current flowing in the tested lead.

Ten measuring coils were prepared for the tests, made from single mode telecommunications fiber of various types (two per each single mode telecommunications fiber standard, one consisting of 40 turns, the other of 80 turns, with a 15 mm turn radius). Measurements were taken for three values of current strengths (True RMS) in the tested wire: 100 A, 300 A, and 500 A.

Instantaneous values of the light polarization angle were measured with the polarimeter. Twenty samples were set for the period of the tested current waveform, equal to 20 ms (the sinusoidal waveform of line-frequency was considered). The time waveforms of the currents being measured are shown in Figure 3, Figure 4 and Figure 5 (for 100 A, 300 A, and 500 A currents, respectively).

The polarimetric current sensor is a “current-to-light polarization angle” (*I*—*α*) converter of chain structure [23]. Its measuring circuit (processing train), shown in Figure 6, is not branched (it has no summation nodes and all processing is done one-way: from the input to the output of the measuring system).

The test of the input signal was based on Fourier analysis. The measuring system shown in Figure 2, includes current clamps (from HEME, model PR1030). The current clamps ware connected to a digital oscilloscope (from RIGOL, model DS1000B) that provided 600 samples per each interval of the tested signal. The measurements were taken for three input waveforms with assumed true RMS current values of 100 A, 300 A, and 500 A.

The number of samples for each of the analyzed waveforms was limited to 20. The samples were taken for specific points in time that corresponded to the sampling performed by the analyzer constituting a part of the polarimeter. Then, complex representations of harmonics in the input waveform were determined, as described by Equation (3), and then, on this basis and based on patterns (6) and (7), the constant component and amplitudes of each harmonic component were determined. The foregoing computation results and Equation (8) were then used to determine the true RMS value of the waveform being analyzed, while Formula (9) yielded the relative harmonic content.

Before proceeding with the DFT analysis of the output signal by the polarimetric current sensor, the instantaneous values of the light polarization angle were converted using Equation (2), taking the Verdet constants for optical waveguides specified in papers [16,19,20].

The final stage of the frequency analysis of the obtained results was the determination of the 20-point DFT input signal and output signal. In the case of the output signal (obtained at the output of polarimetric current sensor with the measuring coil made from single mode telecommunications fiber), the frequency analysis was performed depending on the number of turns of measuring coil, the standard of single mode telecommunications fiber. See Figure 7, Figure 8 and Figure 9 for the results.

The Fourier analysis is a classic apparatus used to analyze signals in most sensors, including polarimetric sensors. Based on the results (Figure 7, Figure 8 and Figure 9) obtained, it can be concluded that regardless of the standard of a single mode telecommunication fiber, for a given number of turns of the measuring coil, the spectral distribution is almost identical. This means that the optical fibers used introduce a measurement error that is similar in magnitude.

An in-depth analysis, using the Equations (8) and (9), allows to determine the total harmonic distortion (THD). Based on the calculations performed, it can be concluded that the considered optical converter does not introduce significant distortions, as evidenced by the value of the harmonic content factor of the signal at its output. On the basis of the obtained results, it is possible to determine the range in which THD is contained, i.e., from 0.53% to 0.73%. The minimum THD value accompanies the standard fiber—G.652, and the maximum value—G.655 for the crown profile (Figure 10).

The obtained results of the frequency analysis allow for determining the measurement errors depending on the standard of optical fiber used and the number of turns of the polarimetric sensor measuring coil. For this purpose, the following definition was used [40]:(10)$$\Delta I=I_{out}-I_{in}$$
(11)$$\delta_{I}=\frac{\Delta I}{I_{out}}\cdot 100\%$$
where: $I_{out}$—true RMS value of the current at the output of the polarimetric current sensor [A] (Figure 11) and $I_{in}$—true RMS value of the current at the input of the polarimetric current sensor [A].

Using the Equations (10) and (11), the absolute and relative error of the measurement were determined (Figure 12 and Figure 13). Based on the obtained results, it can be concluded that the molar concentration of the GeO_2_ impurity in the fiber core (the standard of the single-mode telecommunications fiber used) and the number of turns of the fiber optic coil do not significantly affect the value of the relative error of the current measurement using the *I*—*α* fiber optic converter. The relative processing error ranges from −0.6% to 0.0%. These are very small values, and above all limit values, i.e., those that do not exceed the instrument’s class. Based on the value of the relative borderline error (Figure 13) and the IEC 44-1 [41] standard concerning basic parameters of current transformers, the *I*—*α* converters concerned can be classified in the group of measuring transformers, the class of which depends on the RMS value of the measured current and on the standard of the single mode telecommunications fiber optical used for building the fiber optic measuring coil.

## 5. Conclusions

The aim of this study was to perform a frequency analysis of the polarimetric current sensor with an optical fiber measuring coil made of a single mode telecommunications fiber. This goal resulted directly from the conclusions drawn during the commercialization of the considered sensor and was achieved.

The frequency analysis using the Discrete Fourier Transform (DFT) confirmed that, regardless of the standard of a single mode telecommunications fiber, for a given number of turns of the measuring coil, the spectral distribution is almost identical. Additionally, the low value of the total harmonic distortion (THD) proves that the considered sensor does not introduce additional distortions and distortions of their waveforms when measuring alternating currents.

Based on the presented results, it can be concluded that single mode telecommunication optical fibers, regardless of their standard, can be used to build a measuring coil for polarimetric current sensors. This is confirmed by the fact that the molar concentration of the GeO_2_ in the core of fiber (standard of the single mode telecommunications fiber) and the number of turns of fiber optic measuring coil do not have a significant effect on the value of the total harmonic distortion or the relative error in the measurement of the current using *I*—*α* converter.

The presented results of the Fourier analysis have proved accuracy of the method for designing single mode optical fibers and the method of determining their material properties, described in papers [16,19,20,33,36]. This has been confirmed by the foregoing results of the true RMS value of current measured directly, using the Polarimetric current sensor, which overlap with the results measured with the current clamps and the Rogowski coil.

## Figures and Tables

**Figure 1 sensors-21-03008-f001:**
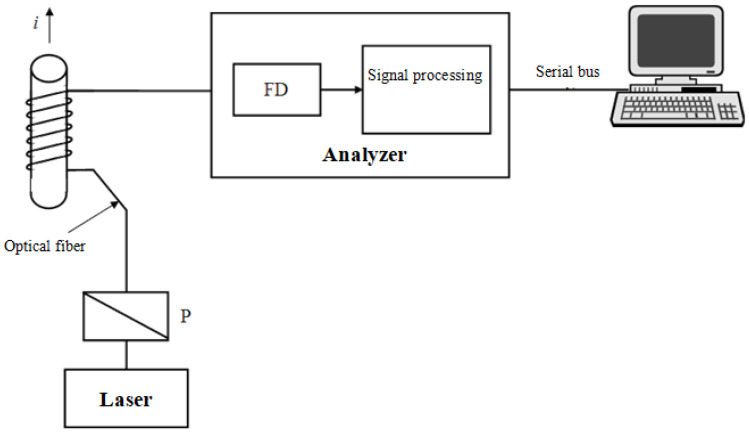
Block diagram of the polarimetric current strength probe with the fiber optic measuring coil: P: polarizer setting the initial and constant polarization; FD: photodetector in the polarization analyzer [19].

**Figure 2 sensors-21-03008-f002:**
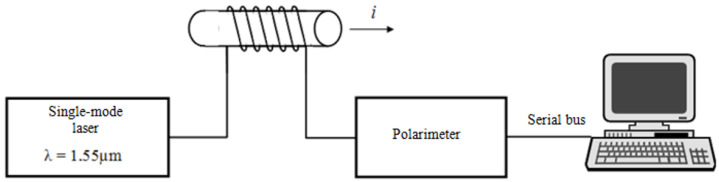
Diagram of the measuring system and the polarimetric current strength probe with the fiber optic measuring coil.

**Figure 3 sensors-21-03008-f003:**
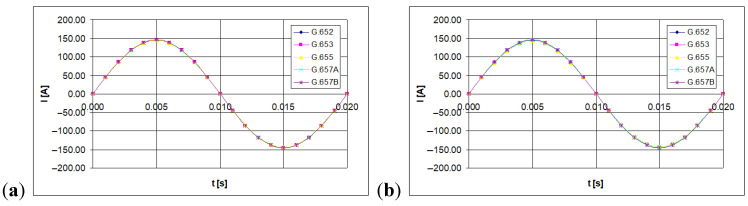
The time waveform measured with forced RMS current value of 100 A, depending on the single mode telecommunications fiber standard [own results]: (**a**) for 40-turns measuring coil and (**b**) for 80-turns measuring coil.

**Figure 4 sensors-21-03008-f004:**
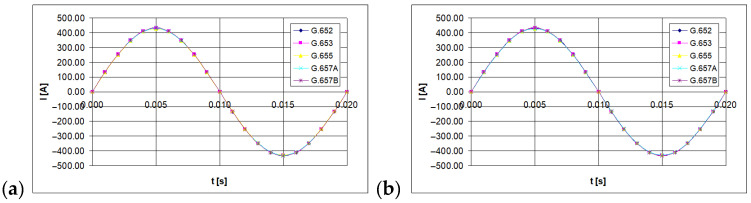
The time waveform measured with forced RMS current value of 300 A, depending on the single mode telecommunications fiber standard [own results]: (**a**) for 40-turns measuring coil and (**b**) for 80-turns measuring coil.

**Figure 5 sensors-21-03008-f005:**
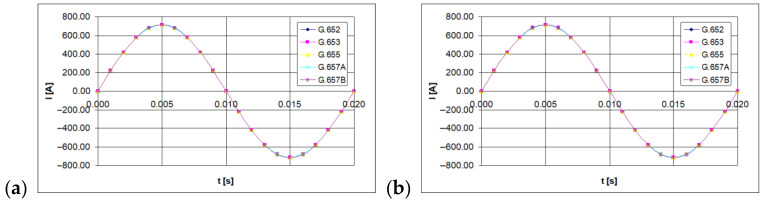
The time waveform measured with forced RMS current value of 500 A, depending on the single mode telecommunications fiber standard [own results]: (**a**) for 40-turns measuring coil and (**b**) for 80-turns measuring coil.

**Figure 6 sensors-21-03008-f006:**
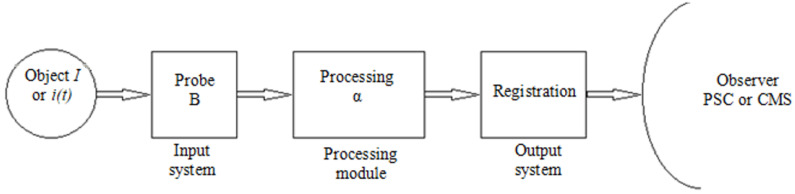
Measuring circuit (processing train) with chain structure for the polarimetric current sensor with the fiber optic measuring coil (Object: power line phase conductor; Probe: optical fiber; Processing: polarimeter; Registration: PC; Observer: the person taking the measurement; PSC: power safety controls; and CMS: control and monitoring systems).

**Figure 7 sensors-21-03008-f007:**
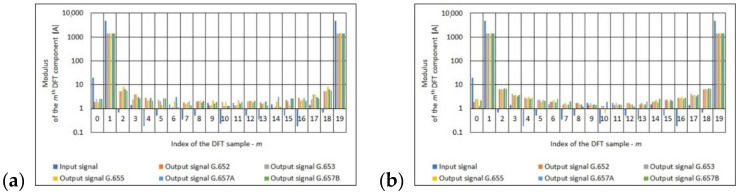
Values of the *m*th DFT input component (100 A True RMS) and output component for the different standards of single mode telecommunications fibers [own results]: (**a**) for 40-turns measuring coil and (**b**) for 80-turns measuring coil.

**Figure 8 sensors-21-03008-f008:**
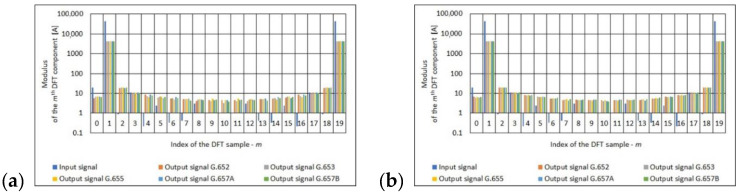
Values of the *m*th DFT input component (300 A True RMS) and output component for the different standards of single mode telecommunications fibers [own results]: (**a**) for 40-turns measuring coil and (**b**) for 80-turns measuring coil.

**Figure 9 sensors-21-03008-f009:**
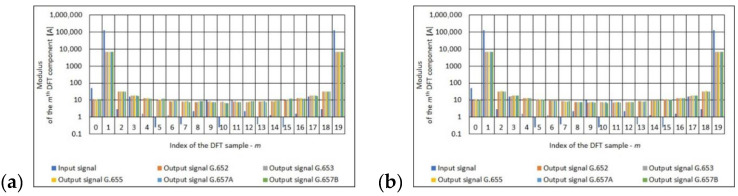
Values of the *m*th DFT input component (500 A True RMS) and output component for the different standards of single mode telecommunications fibers [own results]: (**a**) for 40-turns measuring coil and (**b**) for 80-turns measuring coil.

**Figure 10 sensors-21-03008-f010:**
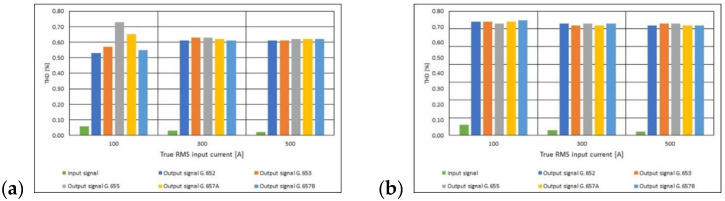
Values of THD for polarimetric current sensor with measuring coil made of the different standards of single mode telecommunications fibers [own results]: (**a**) for 40-turns measuring coil and (**b**) for 80-turns measuring coil.

**Figure 11 sensors-21-03008-f011:**
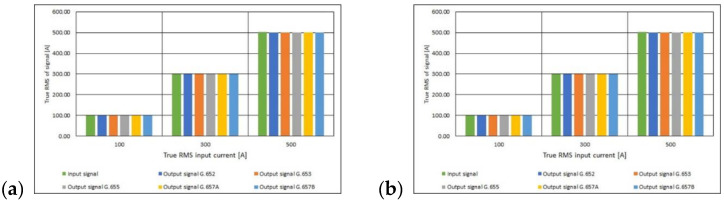
Values of True RMS of measuring current for polarimetric current sensor with measuring coil made of the different standards of single mode telecommunications fibers [own results]: (**a**) for 40-turns measuring coil and (**b**) for 80-turns measuring coil.

**Figure 12 sensors-21-03008-f012:**
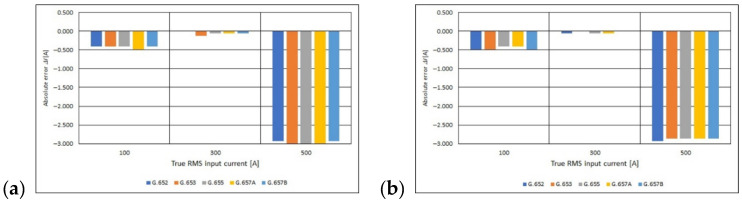
Values of absolute error for polarimetric current sensor with measuring coil made of the different standards of Scheme 40. (**a**) Turns measuring coil and (**b**) for 80-turns measuring coil.

**Figure 13 sensors-21-03008-f013:**
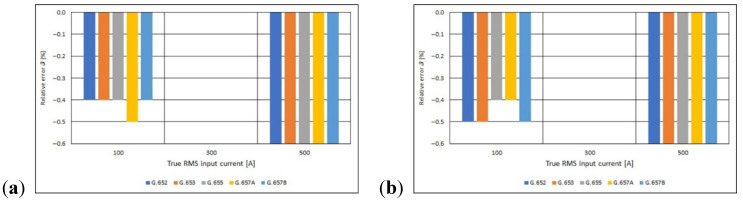
Values of relative error for polarimetric current sensor with measuring coil made of the different standards of single mode telecommunications fibers [own results]: (**a**) for 40-turns measuring coil and (**b**) for 80-turns measuring coil.

## Data Availability

Data sharing is not applicable to this article.

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
