# Peer review of "Testing of a Polarimetric Current Sensor in the Frequency Domain"

_sensors, 2021, doi:10.3390/s21093008_

Round 1
Reviewer 1 Report
Dear colleagues,
your article is about a polarimetric current sensor with signal processing using DFT apparatus.
Obviously, the title of the article misleads the reader a little. In fact, it should be called "Analysis of the spectral composition of the current using a digital Fourier transform apparatus according to the readings of a fiber-optic polarimetric sensor on different types of fibers."
The fact is that there are many developments of current sensors based on the method of reflectometric analysis of the signal of a fiber current sensor in the frequency domain (Google). Namely, in the frequency domain.
In addition, the introduction to the article is very poor and contains only 1 or 2 works of scientists, as correctly noted by authors from around the world for more see Google). A lot of authors' works themselves are cited, and only some general overview publications about sensors and general publications concerning signal DFT analysis, which, however, were not reflected in the formation of a clear statement of the research problem by the authors.
Further sections 2 and 3 are a repetition of the classic descriptions of polarimetric current fiber optic sensors and DFT. The following sections, which, as it were, should reflect the objective function of the article, do not reflect any difference in the use of different types of fibers. Which, in fact, did not raise doubts, since bending losses are considered.
And finally, the conclusion is very general and does not allow us to focus on any significant result of the authors obtained in the course of research. Fourier analysis is a classic apparatus used in most current sensors and not only optical ones, for analyzing the spectral composition of the current flowing in the energy network.
Unfortunately, it is impossible to recommend this article for publication.
Author Response
Thank you for the in-depth review, the detailed comments of which allowed us to improve the prepared article.
- Obviously, the title of the article misleads the reader a little. In fact, it should be called "Analysis of the spectral composition of the current using a digital Fourier transform apparatus according to the readings of a fiber-optic polarimetric sensor on different types of fibers."
The fact is that there are many developments of current sensors based on the method of reflectometric analysis of the signal of a fiber current sensor in the frequency domain (Google). Namely, in the frequency domain.
Response: DFT was used in the work to analyze the operation of the polarimetric sensor. This means that on the example of various designs of the measuring coil and for different current values, the DFT was used to show that when measuring alternating currents, it does not introduce additional distortions and distortions of their waveforms.
The polarimetric sensor does not use the reflectometric method. The reflectometric method examines the return signal resulting from Rayleigh scattering. The polarimetric method analyzes the twist angle of the plane of polarization of light regardless of its power. This means that the bends only affect the length of the measuring coil.
- In addition, the introduction to the article is very poor and contains only 1 or 2 works of scientists, as correctly noted by authors from around the world for more see Google). A lot of authors' works themselves are cited, and only some general overview publications about sensors and general publications concerning signal DFT analysis, which, however, were not reflected in the formation of a clear statement of the research problem by the authors.
Response: The introduction to the article has been improved. Reference was made to other publications mentioning polarimetric sensors. The novelty of the author's concept was indicated, which is the use of single-mode telecommunication optical fibers to build the sensor's measuring coil.
- Further sections 2 and 3 are a repetition of the classic descriptions of polarimetric current fiber optic sensors and DFT. The following sections, which, as it were, should reflect the objective function of the article, do not reflect any difference in the use of different types of fibers. Which, in fact, did not raise doubts, since bending losses are considered.
Response: Chapter 2 was modified by indicating the original concept of using single-mode telecommunication optical fibers in the classic concept of a polarimetric sensor. Additionally, the aspect of the use of commonly available lasers operating at 1310 nm and 1550 nm wavelength was indicated. The third chapter shows which formulas result from verbal considerations contained in the work on DFT.
4. And finally, the conclusion is very general and does not allow us to focus on any significant result of the authors obtained in the course of research. Fourier analysis is a classic apparatus used in most current sensors and not only optical ones, for analyzing the spectral composition of the current flowing in the energy network.
Response: Their discussion is presented immediately after the results of the frequency analysis. As a result, conclusions were remodeled that specifically define the obtained results, which confirm the purpose of this study.
Reviewer 2 Report
The article is interesting; however, I have a few comments to improve the manuscript.
- The Figure 3 to 5 and 7 – 9 needs more elaborate description. The overall outcomes and significance of these results were vaguely stated in this manuscript.
- The overall organization of the article needs to be improved. The results of Figures 10 – 13 were included in the conclusion. It is suggested to add these results in the previous section. The conclusion needs to include the inference of overall results, the significance, limitations, and future work.
Author Response
We would like to thank you for your in-depth review, the detailed comments of which allowed us to improve the prepared article.
- The Figure 3 to 5 and 7 – 9 needs more elaborate description. The overall outcomes and significance of these results were vaguely stated in this manuscript.
Response: The graphical results of the frequency analysis are discussed in detail.
As a result, conclusions were remodeled that specifically define the obtained results, which confirm the purpose of this study.
- The overall organization of the article needs to be improved. The results of Figures 10 – 13 were included in the conclusion. It is suggested to add these results in the previous section. The conclusion needs to include the inference of overall results, the significance, limitations, and future work.
Response: The introduction to the article has been improved. Reference was made to other publications mentioning polarimetric sensors.
The novelty of the author's concept was indicated, which is the use of single-mode telecommunication optical fibers to build the sensor's measuring coil.
Reviewer 3 Report
The paper describes the characterisation of polarimetric current sensors in which the measuring coil is made of various single mode optical fibres which meet telecommunications standards. The characterisation covers studies of both time-domain and frequency domain signals, the latter being obtain by use of a Discrete Fourier Transform. The work described is in my view incremental in nature; by the authors own admission it proves the accuracy of the method described in their references [10,11,12,13,18]. Nevertheless, the frequency analysis and study of the “current-to-light polarization angle” presented for various fibres is of some interest.
Before possible publication, the paper needs significant revision. Specifically the authors need to bring out the novel contributions made and their importance. Its present structure makes it unsuitable for publication without such revisions.
I have the following specific comments, written in the order in which the reader comes across them when reading the paper.
Abstract. Clarify what the key result(s) presented are. The last sentence is very long and is an example of where benefit could be gained from tidying up the English.
Introduction. The background material is interesting, but it does not tell the reader clearly enough what the novel contribution of the paper is. (It appears to be to fill the gap in the literature by using standard fibres operating at wavelengths of 1310 and 1550 nm. The conclusions talk about commercialisation of a measurement system based on such fibres which is not mentioned at all here.) A few additional sentences would help enormously.
Section 3. What is added over author reference [15]? State clearly what is new, and why/where it will be used in the analysis of results.
Section 4
Avoid section title appearing immediately before the page break in the final version.
Line 199. ‘Supposed to be’ or ‘was’? If it wasn’t, why not?
Fig 3 etc – these are all your own results, the caption could be read that it is just those in part b which are yours.
There is no discussion of results – until the conclusions! This must be rectified. See further comments below.
Conclusions
Conclusions should present only conclusions, not new discussion. In fact the present Conclusions presents new figures (analysis). A separate discussion section is needed, or at least discussion of results added to section 4. For example lines 295- is not a conclusion but new analysis.
If the analysis and discussion were to be separated in a significantly revised version of the paper this will then enable the authors to draw very definite conclusions – which map Abstract and Introduction – and enable the claimed novelty (e.g. text starting around line 291) to be seen much more clearly by the reader.
Author Response
We would like to thank you for your in-depth review, the detailed comments of which allowed us to improve the prepared article.
- Abstract. Clarify what the key result(s) presented are. The last sentence is very long and is an example of where benefit could be gained from tidying up the English.
Response: The abstract of the article has been modified in such a way as to highlight the key results, which are of particular importance for the work.
- Introduction. The background material is interesting, but it does not tell the reader clearly enough what the novel contribution of the paper is. (It appears to be to fill the gap in the literature by using standard fibres operating at wavelengths of 1310 and 1550 nm. The conclusions talk about commercialisation of a measurement system based on such fibres which is not mentioned at all here.) A few additional sentences would help enormously.
Response: The introduction to the article has been improved. Reference was made to other publications mentioning polarimetric sensors. The novelty of the author's concept was indicated, which is the use of single-mode telecommunication optical fibers to build the sensor's measuring coil.
- Section 3. What is added over author reference [15]? State clearly what is new, and why/where it will be used in the analysis of results.
Response: Chapter 2 was modified by indicating the original concept of using single-mode telecommunication optical fibers in the classic concept of a polarimetric sensor. Additionally, the aspect of the use of commonly available lasers operating at 1310 nm and 1550 nm wavelength was indicated. The third chapter shows which formulas result from verbal considerations contained in the work on DFT.
- Section 4
Avoid section title appearing immediately before the page break in the final version.
Line 199. ‘Supposed to be’ or ‘was’? If it wasn’t, why not?
Fig 3 etc – these are all your own results, the caption could be read that it is just those in part b which are yours.
Response: Chapter 4 has been modified so that the calibration wording is not ambiguous.
- There is no discussion of results – until the conclusions! This must be rectified. See further comments below.
Response: Additionally, their discussion is provided immediately after the results of the frequency analysis.
- Conclusions
Conclusions should present only conclusions, not new discussion. In fact the present Conclusions presents new figures (analysis). A separate discussion section is needed, or at least discussion of results added to section 4. For example lines 295- is not a conclusion but new analysis.
If the analysis and discussion were to be separated in a significantly revised version of the paper this will then enable the authors to draw very definite conclusions – which map Abstract and Introduction – and enable the claimed novelty (e.g. text starting around line 291) to be seen much more clearly by the reader
Response: As a result, conclusions were remodeled that specifically define the obtained results, which confirm the purpose of this study. The graphical results of the frequency analysis are discussed in detail.
Round 2
Reviewer 3 Report
I thank the authors for addressing my concerns. They have done this to my satisfaction and I believe that the paper can now be published.
Before the paper appears in print I wold suggest two minor editorial improvements to the sections added:
In lines 306-307 'This means that the optical fibers used introduce a measurement error similar to the value' would read better as 'This means that the optical fibers used introduce a measurement error that is similar in magnitude.'
Lines 351-353 (First lines of the Conclusions) are also rather convoluted.
Author Response
Thank you kindly for your review. Valuable comments made the work more precise and clear for the reader. After the changes to lines 307 to 308, the sentence becomes: "This means that the optical fibers used introduce a measurement error that is simi-lar in magnitude." After the changes on lines 352 to 355, the introduction to the conclusions takes the form: "The aim of this study was to perform a frequency analysis of the polarimetric cur-rent sensor with an optical fiber measuring coil made of a single mode telecommunica-tions fiber. This goal resulted directly from the conclusions drawn during the commer-cialization of the considered sensor and was achieved."
This manuscript is a resubmission of an earlier submission. The following is a list of the peer review reports and author responses from that submission.